# Enhancing Compositional Generalization via Compositional Feature Alignment

**Haoxiang Wang**                                                    *hwang264@illinois.edu*
*Department of Electrical and Computer Engineering*
*University of Illinois Urbana-Champaign*

**Haozhe Si**                                                        *haozhes3@illinois.edu*
*Department of Electrical and Computer Engineering*
*University of Illinois Urbana-Champaign*

**Huajie Shao**                                                      *hshao@wm.edu*
*Department of Computer Science*
*William and Mary*

**Han Zhao**                                                         *hanzhao@illinois.edu*
*Department of Computer Science*
*University of Illinois Urbana-Champaign*

**Reviewed on OpenReview:** *https://openreview.net/forum?id=k3d5COYvfK*

## Abstract

Real-world applications of machine learning models often confront data distribution shifts, wherein discrepancies exist between the training and test data distributions. In the common multi-domain multi-class setup, as the number of classes and domains scales up, it becomes infeasible to gather training data for every domain-class combination. This challenge naturally leads the quest for models with Compositional Generalization (CG) ability, where models can generalize to unseen domain-class combinations. To delve into the CG challenge, we develop CG-Bench, a suite of CG benchmarks derived from existing real-world image datasets, and observe that the prevalent pretraining-finetuning paradigm on foundational models, such as CLIP and DINOv2, struggles with the challenge. To address this challenge, we propose Compositional Feature Alignment (CFA), a simple two-stage fine-tuning technique that i) learns two orthogonal linear heads on a pretrained encoder with respect to class and domain labels, and ii) fine-tunes the encoder with the newly learned head frozen. We theoretically and empirically justify that CFA encourages compositional feature learning of pretrained models. We further conduct extensive experiments on CG-Bench for CLIP and DINOv2, two powerful pretrained vision foundation models. Experiment results show that CFA outperforms common finetuning techniques in compositional generalization, corroborating CFA's efficacy in compositional feature learning. The code is released at https://github.com/Haoxiang-Wang/Compositional-Feature-Alignment.

## 1 Introduction

Over the past decade, machine learning has emerged as a transformative technology, driving advancements across various domains such as computer vision (He et al., 2016a;b), natural language processing (Devlin et al., 2019; Brown et al., 2020), biology (Jumper et al., 2021), etc. These innovations have been fueled by the development of increasingly sophisticated models, the availability of large-scale datasets, and the growth of computational power. However, a crucial obstacle persists in applying machine learning models to real-world scenarios: their performance tends to degrade significantly when confronted with data

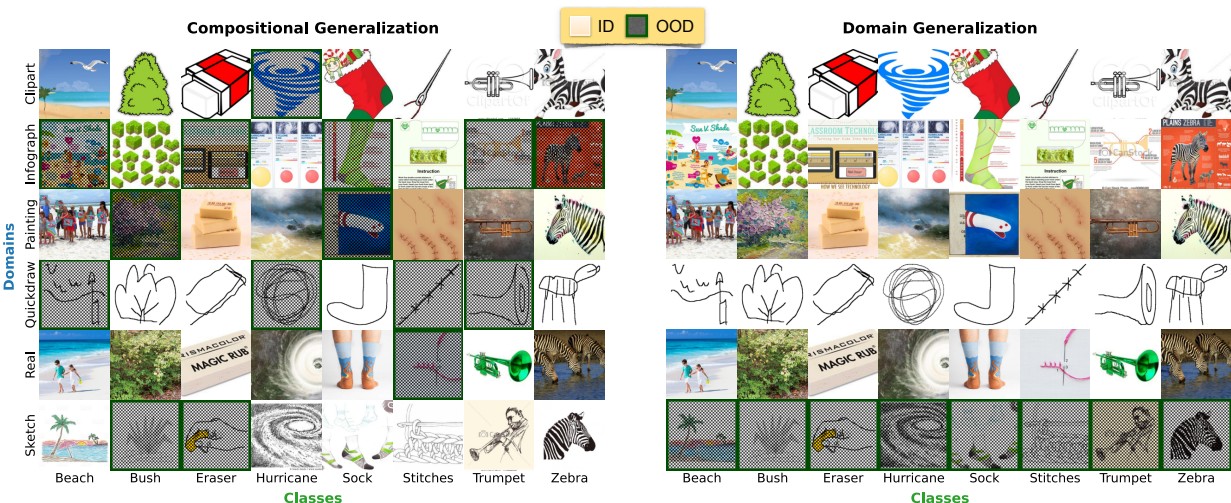

Figure 1: Compositional generalization (CG) vs. domain generalization (DG). Masked entries are unseen domain-class combinations, while unmasked ones exist in the training dataset.

distribution shifts (Koh et al., 2021; Gulrajani & Lopez-Paz, 2021; Santurkar et al., 2021), where the data distribution during testing differs from that used in training.

In an effort to overcome this problem, the machine learning research community has turned its attention to Out-of-Distribution (OOD) generalization, with the goal of developing models that are robust under data distribution shifts. Existing research primarily investigates various types of data distribution shifts, such as domain generalization (Gulrajani & Lopez-Paz, 2021; Koh et al., 2021), subpopulation shift (Santurkar et al., 2021; Yang et al., 2023), input corruption (Deng et al., 2009), and spurious correlation (Sagawa et al., 2020). While generalizing to these different type of distribution shifts has garnered significant attention, there exists another realistic yet understudied challenge in OOD generalization: *compositional generalization* (CG).

Within the multi-domain, multi-class context, assumes we have $E$ domains (i.e., environments) and $K$ classes, leading to $E \times K$ pairs of domain and class combinations, which could be formulated as elements in a $E \times K$ matrix as shown in Figure 1. In domain generalization (DG), the learner has access to data from all the classes and all the domains and aims to make predictions on data from a new, unseen domain. Nonetheless, in real-world scenarios, given the large number of categories, e.g., 1000 classes in ImageNet, one cannot always collect the complete data from all the domains. Put in another word, the training data might not cover all the possible combinations of domains and classes, as represented by each cell in the matrix in Figure 1. This sparsity pattern becomes especially pronounced when the number of classes or environments is large because collecting comprehensive training data for each combination becomes a formidable task. In such cases, a key challenge arises: *can the model generalize to unseen domain-class combinations?* This is the compositional generalization (CG) challenge we aim to tackle in this work.

The CG challenge manifests ubiquitously across various real-world applications. For example, data distribution shifts in certain existing DG datasets, such as iWildCam (Beery et al., 2020; Koh et al., 2021), are more accurately characterized by CG than DG. Moreover, we find that the widely-used method of finetuning pretrained (foundation) models struggles to tackle the CG challenge. This emphasizes the need for the machine learning community to recognize and address this emerging distribution shift challenge with innovative solutions.

**Our Contributions** In our attempt to tackle this challenge, we draw inspiration from existing lines of research such as invariant risk minimization (IRM) (Arjovsky et al., 2019) and invariant-feature subspace recovery (ISR) (Wang et al., 2022). In particular, Wang et al. (2022) showed that under certain structural conditions in the data generative process, post-processing methods via subspace projection can effectively learn invariant features that can generalize across unseen domains from the same data generative process but under different interventions on non-causal factors. Empirically, we find that if the learned features (i.e., the outputs of the last hidden layer) conform to a compositional structure where the subspace of *domain-*

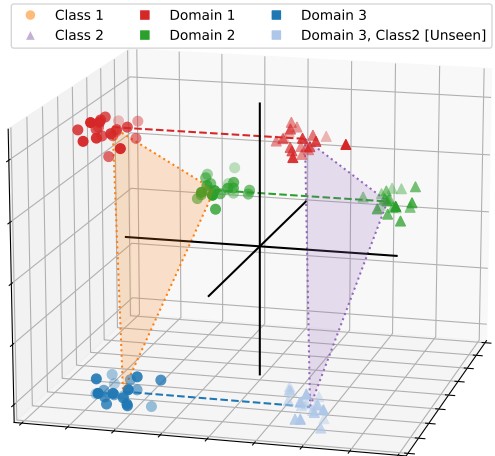

Figure 2: Illustration of a desired compositional feature structure for compositional generalization.

*related* features is orthogonal to that of *class-related* features, the corresponding model can generalize across unknown domain-class pairs. Motivated by this observation, to induce features that match this compositional structure, we introduce a two-stage finetuning approach termed Compositional Feature Alignment (CFA), which is also inspired by recent progress in the literature of neural collapse (Papyan et al., 2020; Zhu et al., 2021; Yang et al., 2022).

More specifically, upon the features given by the encoder, we construct two heads, one for predicting the target label of interest and the other for predicting the domain index. Note that the two-head architecture is not new, and has been widely used in domain adversarial neural networks (Ganin & Lempitsky, 2015; Zhao et al., 2018). However, different from domain adversarial neural networks where adversarial training through minimax optimization is needed, our proposed method is computationally lightweight and can be divided into two stages.

CFA first identifies a proper compositional feature structure via a two-head regularized linear probing (i.e., training linear heads with the encoder frozen). Subsequently, the encoder undergoes finetuning with the heads being frozen. Leveraging tools from the neural collapse literature, we theoretically prove that CFA can effectively align features with the compositional feature structure under mild assumptions. Furthermore, we construct a synthetic Color-CIFAR dataset to examine CFA empirically and observe that CFA can indeed align features with the desired compositional feature structure.

To facilitate the studies of compositional generalization, we curate a suite of benchmarks for the compositional generalization challenge, building on four real-world image datasets: OfficeHome (Venkateswara et al., 2017), DomainNet (Peng et al., 2019), WILDS-iWildCam (Beery et al., 2020), and WILDS-FMoW (Christie et al., 2018). We consider two powerful pretrained vision encoders, CLIP (Radford et al., 2021) and DINOv2 (Oquab et al., 2023), with ViT-B (Dosovitskiy et al., 2021) architecture, and apply different finetuning methods to them, including linear probing, full finetuning, LP-FT (Kumar et al., 2022), reweighting and our proposed CFA. Extensive experimental results on CG-Suite show that CFA-finetuned models can indeed generalize to unseen domain-class combinations better than other finetuning methods. We hope that the curated CG-Suite can facilitate future research on compositional generalization.

## 2 Compositional Feature Alignment

The key to the CG challenge is to identify and encode the compositional relationship between classes and domains by learning the features. Hence, it is important to first understand what kind of feature structures are desired for compositional generalization. To this end, we first provide a formal definition of the compositional feature structure, and then explain our motivations behind the definition.

**Definition 1** (Compositional Feature Structure)**.** *For any input $x$ from class $y \in \{1, \ldots, K\}$ and domain $e \in \{1, \ldots, E\}$, its feature $z \in \mathbb{R}^d$ satisfies the compositional feature structure as long as $z$ can be decomposed as:*

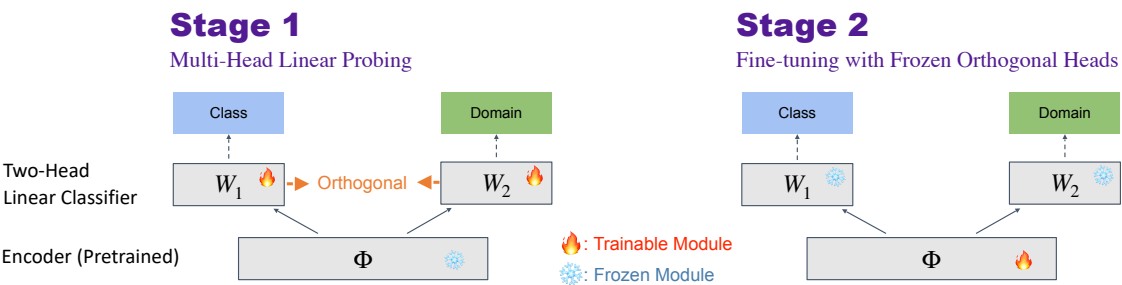

Figure 3: Illustration of our proposed method, Compositional Feature Alignment (CFA).

$$\text{Class Feature: } z_1 \sim \mathcal{N}(\mu_1^y, \Sigma_1^y) \in \mathbb{R}^{d_1},$$
$$\text{Domain Feature: } z_2 \sim \mathcal{N}(\mu_2^e, \Sigma_2^e) \in \mathbb{R}^{d_2}$$

$$\text{Total Feature: } z = R \begin{bmatrix} z_1 \\ z_2 \\ z_{\text{noise}} \end{bmatrix}$$

where $z_{\text{noise}} \in \mathbb{R}^{d-d_1-d_2}$ represent noise features irrelevant to classes and domains, and $R \in \mathbb{R}^{d \times d}$ is a full-rank orthonormal matrix. Note that $\mu_1^y, \Sigma_1^y$ have dependence on $y$, while $\mu_2^e, \Sigma_2^e$ rely on $e$.

Fig. 2 provides a visualization of this compositional feature structure for a simple setup of 2 classes and 3 domains, where one domain-class combination is absent in the training set. It is evident that class features and domain features exist in orthogonal subspaces, as required by Definition 1. In this case, a linear classifier that exclusively utilizes class features and disregards domain and noise features can effectively generalize the unseen domain-class combination.

It is noteworthy that even with the perfect alignment of learned features to this compositional structure on all training data, there is no guarantee that features from unseen domain-class combinations will still conform to this structure.

## 2.1 Method: Training with Frozen Orthogonal Heads under Normalization

Though we have defined an ideal feature structure for compositional generalization, the neural features produced by pretrained models may not align with this structure. In this section we introduce our method to encourage the learned features to follow the above structure. At a high level, the proposed method contains two stages of finetuning from pretrained models. We include an auxiliary linear head for domain prediction, complementing the pre-existing class prediction head. The first stage involves multi-label linear probing, which adjusts the two heads to achieve an optimal compositional feature structure. In the second stage, we fine-tune the encoder, while keeping the two heads frozen. The final product is a finetuned encoder that generates features in alignment with the predetermined compositional feature structure from stage one. The two stages, diagrammatically represented in Fig. 3, are detailed below, followed by a discussion on our rationale behind the algorithm design.

**Stage 1** (Multi-Label Linear Probing). *We begin with a pretrained encoder, $\Phi(\cdot)$ that maps inputs to $d$-dimensional features of unit norm, where $d > K + E$). We construct two linear heads without bias terms, denoted by $W_1 \in \mathbb{R}^{K \times d}$ and $W_2 \in \mathbb{R}^{E \times d}$. Keeping $\Phi(\cdot)$ frozen, we train these heads with two cross-entropy loss terms, which take into account both class and domain labels. An orthogonality constraint ensures $W_1$ and $W_2$ span orthogonal subspaces.*

*Mathematically, the optimization objective of the first stage can be written as*

$$\min_{W_1, W_2} \frac{1}{N} \sum_{(x,y,e) \in D_{\text{train}}} \frac{1}{K} \ell_{\text{CE}}(\beta_1 \cdot W_1 \Phi(x), \ y) + \lambda \frac{1}{E} \ell_{\text{CE}}(\beta_2 \cdot W_2 \Phi(x), \ e) \tag{1}$$

$$\text{s.t. } \beta_1, \beta_2, \lambda > 0 \text{ and } W_1 \in \mathcal{U}(d)^K, W_2 \in \mathcal{U}(d)^E, W_1 W_2^{\mathsf{T}} = \mathbf{0} \tag{2}$$

where $D_{\text{train}}$ represents the training set, $\ell_{\text{CE}}$ is the cross-entropy loss, $\beta_1, \beta_2$ are inverse temperature parameters (also called as logit scale in CLIP (Radford et al., 2021)), $\mathcal{U}(d)$ denotes the set of $d$-dimensional unit vectors, and $\mathbf{0}$ stands for the zero matrix.

**Stage 2** (finetuning with Frozen Heads). *We then freeze the trained $W_1$ and $W_2$, and proceed to fine-tune the encoder $\Phi(\cdot)$ end-to-end, using the same multi-label cross-entropy loss function.*

*The optimization objective of this finetuning stage can be expressed as*

$$\min_{\Phi} \frac{1}{N} \sum_{(x,y,e)\in\mathcal{D}} \frac{1}{K}\ell_{\mathrm{CE}}(\beta_1 \cdot W_1\Phi(x), y) + \lambda\frac{1}{E}\ell_{\mathrm{CE}}(\beta_2 \cdot W_2\Phi(x), e) \tag{3}$$

The following discussion explains our motivations and reasoning underlying the algorithm design:

- ***Freezing Head for Feature Alignment***: Recent work on neural collapse indicate that during the training of a multi-class classifier using cross-entropy loss, freezing the linear head according to a simplex structure can guide the features to align with the frozen head (Zhu et al., 2021; Yang et al., 2022). This observation implies that the features of data in class $y$ to collapse in the direction of the row vector of the classifier corresponding to class $y$. In addition, we empirically observe that the head-freezing technique does not compromise the model's performance compared to end-to-end finetuning. We include the details regarding this experiment in an ablation study in Appendix C. Inspired by these, we devise the two-stage strategy where the Stage 1 determines the optimal head weights for our compositional feature structure, and the Stage 2 finetunes the encoder with frozen head weights to align the features with the feature structure.

- ***Linear Probing Two Orthogonal Heads***: Unlike research work on neural collapse that focus exclusively on class prediction (Papyan et al., 2020; Zhu et al., 2021; Yang et al., 2022), our work also accounts for the effects of domains, as outlined in Definition 1. We therefore introduce an auxiliary head, $W_2$, for domain prediction, alongside the original class prediction head denoted as $W_1$. Thus, for a sample $x$, the encoder, $W_1$, and $W_2$ predict the class and domain labels based on the feature $\Phi(x)$. Definition 1 implicitly poses an orthogonality requirement on domain-related and class-related features since $R$ is an orthonormal matrix. To meet this feature orthogonality requirement, we impose an orthogonality constraint on the two heads (i.e., $W_1W_2^{\mathsf{T}} = \mathbf{0}$).

- ***Normalizing Features and Weights to Address Data Imbalance***: While Zhu et al. (2021); Yang et al. (2022) provide a theoretical justification for head freezing, their theory assumes class-balanced training data. In the case of data imbalance, Thrampoulidis et al. (2022) shows that the head and features may become misaligned. Upon reviewing the technical details of Thrampoulidis et al. (2022), we find that this misalignment can be rectified by normalizing features and head weights to a hyper-sphere. This normalization ensures constant norms for features and head weights, thereby ensuring alignment (cf. Theorem 1). Consequently, we assume that the features produced by the encoder $\Phi$ are also normalized to unit norm, which is a common practice in modern vision model pretraining such as CLIP (Radford et al., 2021), SimCLR (Chen et al., 2020), MoCo (He et al., 2020), DINO (Caron et al., 2021). Additionally, we impose the head normalization constraint $W_1 \in \mathcal{U}(d)^K, W_2 \in \mathcal{U}(d)^E$ in (2), a technique already employed in CLIP (Radford et al., 2021)[1].

## 2.2 Theoretical Guarantee

In the algorithm above, Stage 1 is relatively simple, comprising a joint minimization problem over two linear heads. In contrast, Stage 2 is more complex, as it optimizes a neural encoder using two heads under two cross-entropy loss terms. We offer theoretical justification for Stage 2 below, demonstrating that the finetuned encoder can indeed align features with the two frozen orthogonal heads produced by Stage 1, thereby creating a feature structure that meets the requirement in Definition 1.

In line with recent research on neural collapse (Mixon et al., 2020; Fang et al., 2021; Zhu et al., 2021; Thrampoulidis et al., 2022), we adopt the unconstrained feature model (UFM) or layer-peeled model, where $z = \Phi(x)$ is treated as a free optimization variable in $\mathbb{R}^d$ for every input $x$.

We denote $\mathbf{Z} = [\phi(x_1), \dots, \phi(x_n)] \in \mathbb{R}^{d\times N}$, $\mathbf{Y} = [y_1, \dots, y_N]$, and $\mathbf{E} = [e_1, \dots, e_N]$ as the stack of features, class labels, and environment labels, respectively. In the context of the unconstrained feature model, the

---

[1]Recent studies show that weight normalization for the linear head (without bias) can enhance the performance of fine-tuned CLIP (Goyal et al., 2022; Wang et al., 2023).

optimization objective of Stage 2 is transformed to:

$$\min_{\mathbf{Z}} \frac{1}{KN} \ell_{\mathrm{CE}}(\beta_1 \cdot W_1 \mathbf{Z}, \mathbf{Y}) + \lambda \frac{1}{EN} \ell_{\mathrm{CE}}(\beta_2 \cdot W_2 \mathbf{Z}, \mathbf{E}) \quad \text{s.t.} \quad \mathbf{Z} \in \mathcal{U}(d)^N \tag{4}$$

**Theorem 1** (Feature Alignment). *Assuming the feature dimension $d$ is no smaller than $K + E$, and training data exists for each class and domain (though not necessarily for each domain-class combination), and $W_1$ and $W_2$ are normalized and span orthogonal subspaces such that $W_1 \in \mathcal{U}(d)^K, W_2 \in \mathcal{U}(d)^E$ and $W_1 W_2^{\mathsf{T}} = \mathbf{0}$. Additionally, we assume $\beta_1, \beta_2$ are sufficiently large. The global minimum of* (4) *results in the following: for any $i \in [N]$, denote $z_i$ as the $i$-th column vector of $\mathbf{Z}$, we have*

$$z_i^* = W_1^{\mathsf{T}} \boldsymbol{a}_{y_i} + W_2^{\mathsf{T}} \boldsymbol{b}_{e_i} \tag{5}$$

*where $\boldsymbol{a}_{y_i} \in \mathbb{R}^K$ is a vector depending on the class label $y_i$, and $\boldsymbol{b}_{e_i} \in \mathbb{R}^E$ is a vector relying on the domain label $e_i$.*

This theorem, intuitively, demonstrates that upon optimizing (4) to a global minimum, for any training sample from class $y$ in environment $e$, its corresponding feature $z^*$ can be decomposed as a linear combination of two vectors depending on $y$ and $e$, respectively, and the two vectors live in orthogonal feature subspaces. This indicates that the learned features conform to a compositional feature structure satisfying Definition 1. The complete proof is found in Appendix B, where we leverage theoretical tools from Thrampoulidis et al. (2022) in the proof.

## 3 Empirical Studies

### 3.1 Benchmark Development of CG-Bench

We create CG-Bench, a compositional generalization benchmark built on four datasets previously designed for DG research: Office-Home (Venkateswara et al., 2017), DomainNet (Peng et al., 2019), and iWildCam (Beery et al., 2020) & FMoW (Christie et al., 2018) from the WILDS benchmark (Koh et al., 2021). These datasets span a wide range of application scenarios from common object recognition in web-crawled images to wildlife recognition from camera traps and building recognition from satellite images. Due to the page limit, we elaborate on the motivation for creating the CG-bench and the curation procedure by taking DomainNet as an example. Additional details regarding benchmark curation can be found in Appendix A.

**DomainNet (Peng et al., 2019)** consists of objects in different art styles. In the DomainNet dataset, there are $K = 345$ classes and $E = 6$ domains: {`Clipart`, `Infograph`, `Painting`, `Quickdraw`, `RealImage`, `Sketch`}. In addressing the DG challenge, prior research using DomainNet typically employed a leave-one-out cross-validation strategy. This involves training the model on data from five of the domains and subsequently evaluating its performance on the sixth, omitted domain. In addition to the DG task, it's worth noting that the CG challenge is intrinsically present within DomainNet. To underscore this point, we carried out a preliminary experiment using CLIP.

**CG Challenge in DomainNet** We randomly divide DomianNet into training and evaluation sets, with an 80:20 split. A CLIP model is fully fine-tuned on this training data, and evaluated on validation data from all domain-class combination. We also gathered zero-shot accuracy of the CLIP model for comparison. As a final step, we examined the test accuracies for each domain-class combination, correlating them with the count of their respective training data samples. We only focus on *hard* domain-class combinations that the zero-shot accuracy is below 30%, and visualize evaluation results over these combinations inFigure 4a. Firstly, we notice that certain domain-class combinations possess minimal or even no training samples (e.g., some combinations have only 2 images and neither of them is sampled into the training set). This observation aligns with our considered CG scenario. Within this CG context, both the fine-tuned and zero-shot models encounter difficulties in achieving high test accuracy when the training data for a specific domain-class combination is insufficient. This leads us to conclude that the CG challenge is inherently present in DomainNet, and current zero-shot and fine-tuned models fail to address. Consequently, we are motivated to establish a benchmark for a methodical investigation of this challenge.

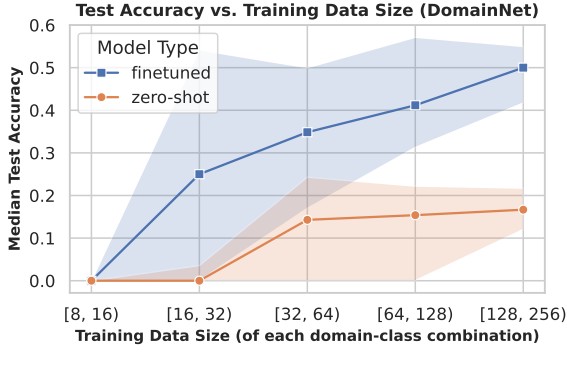

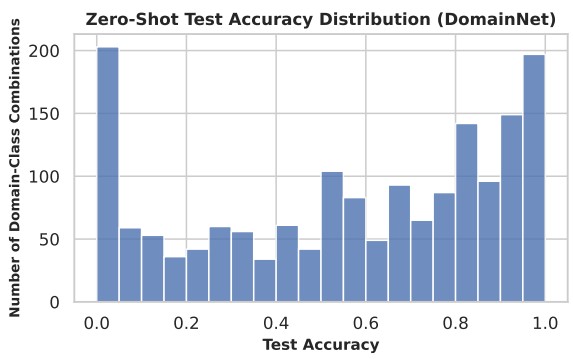

(a) Test Accuracy vs. Training Data Size

(b) Zero-shot Test Accuracy Distribution

Figure 4: Test accuracy statistics for each domain-class combination on DomainNet dataset. **Left**: The test accuracy compared with the number of training data. Points are median accuracy while the shading area is bounded by 25% and 75% quantiles. **Right**: The number of domain-class combinations at different zero-shot test accuracies.

**Benchmark Curation Setup**   Consider data belonging to $E$ domains (i.e., environments) and $K$ classes, resulting in an $E \times K$ matrix of domain-class combinations (such as the demo shown in Fig. 1). A binary mask $M_{\mathrm{id}} \in \{0,1\}^{E \times K}$ is applied to this matrix to indicate *in-distribution* domain-class combinations, ensuring that each row or column contains both 0 and 1 so that the training data includes all classes and domains, while some domain-class combinations may be absent. The complementary binary mask ($M_{\mathrm{ood}} = 1 - M_{\mathrm{id}}$) represents OOD domain-class combinations.

**CG Curation of DomainNet**   We form a $E \times K$ class-environment matrix for DomainNet and evaluate the zero-shot accuracy for every domain-class combination. The resulting data distribution is visualized in Figure 4b. We designate the combinations that fall within the lowest 20% of zero-shot accuracies as the out-of-distribution (OOD) set, while the top 80% constitute the in-distribution (ID) set. To ensure comprehensive representation, each row/column contains at least one entry from the ID set. The ID data is then further segregated into a training set and an ID validation set at a 9:1 ratio. Meanwhile, the OOD data is divided between OOD validation and test sets. When evaluating a model trained on the main training dataset, we assess its performance across the ID validation, OOD validation, and OOD test subsets. For this benchmark, our key metric is the average top-1 accuracy for both the ID and OOD sets.

## 3.2   Evaluations

**Baselines** We take the OpenAI's ViT-B/16 CLIP (Radford et al., 2021) and the Meta's ViT-B/14 DINOv2 (Oquab et al., 2023) as the pretrained model for each benchmark and implement three finetuning strategies as the baseline: **full finetuning**, **linear probing then full finetuning** (*LP-FT*) and **reweighting**. LP-FT (Kumar et al., 2022) is a simple two-stage finetuning strategy that addresses the problem that full finetuning can distort pretrained features and often underperforms linear-probing on out-of-distribution data. *Note:* Our proposed approach bears similarities with LP-FT (Kumar et al., 2022), as both methods start with a linear probing stage and then proceed with finetuning. However, two critical differences are: i) our approach employs two heads with a multi-label cross-entropy loss and imposes an orthogonality constraint during the linear probing stage, and ii) we keep the heads frozen during the second finetuning stage. *Reweighting* (Buda et al., 2018) balances the number of samples from each group in each batch during training. It is robust to group shifts in OOD generalization tasks. For the CG benchmarks, we implemented two versions of reweighting strategies: Reweight-E, which does re-sampling according to the domain labels, and Reweight-Y×E, which balances according to the domain-class combinations.

Table 1: Test accuracy(%) and F1-macro scores (%) of different methods on CG-Bench. Bold values mark the highest accuracy or F1 score. The OOD accuracy of FMoW is the worst region accuracy. The highest accuracy values and those within a range of 0.2 are in **bold**.

| Model | Methods | OfficeHome | | DomainNet | | iWildCam | | | | FMoW | |
|---|---|---|---|---|---|---|---|---|---|---|---|
| | | ID Acc | OOD Acc | ID Acc | OOD Acc | ID Acc | OOD Acc | ID F1 | OOD F1 | ID Acc | OOD Acc |
| CLIP | Zero-Shot | 89.2 | 50.3 | 61.7 | 6.6 | 13.7 | 6.9 | 11.7 | 9.2 | 20.4 | 18.8 |
| | Linear Probing | 90.9 | 41.0 | 72.7 | 4.7 | 72.5 | 14.4 | 42.1 | 22.5 | 37.7 | 27.6 |
| | Fine-Tuning | **94.3** | 51.0 | **82.0** | 7.5 | 74.5 | 16.5 | 43.8 | 22.2 | **65.8** | 38.7 |
| | Fine-Tuning (WiSE) | 93.7 | 52.5 | 76.4 | 8.7 | 67.0 | 13.7 | 31.6 | 17.0 | 49.5 | 40.6 |
| | LP-FT | 93.5 | 43.9 | 81.5 | 5.3 | 74.0 | 17.0 | 42.5 | 26.6 | **65.9** | 40.2 |
| | LP-FT (WiSE) | 93.0 | 42.8 | 79.4 | 5.3 | 74.4 | 18.2 | 44.5 | 28.7 | 56.6 | 36.3 |
| | Reweight-E | 94.0 | 51.9 | 81.2 | 7.4 | **75.3** | 17.2 | 45.2 | 24.3 | 62.4 | **41.8** |
| | Reweight-E (WiSE) | 93.6 | 53.1 | 75.9 | 8.5 | 68.5 | 13.7 | 32.9 | 18.0 | 46.8 | **41.7** |
| | Reweight-Y×E | 93.7 | 52.2 | 81.0 | 7.6 | 72.2 | 17.4 | 41.6 | 30.0 | 58.0 | 41.1 |
| | Reweight-Y×E (WiSE) | 93.4 | 53.4 | 75.5 | 8.5 | 55.9 | 15.1 | 29 | 22.7 | 42.1 | 37.5 |
| | CFA | **94.3** | 54.3 | 81.6 | 7.3 | 74.0 | 18.3 | 43.6 | 31.0 | 65.3 | **41.6** |
| | CFA (WiSE) | 93.1 | **56.9** | 76.5 | **9.2** | 74.6 | **19.7** | **45.6** | **32.5** | 53.5 | 36.6 |
| DINOv2 | Fine-Tuning | 91.8 | 38.6 | **82.4** | 5.3 | 76.4 | 14.4 | 47.6 | 18.3 | 66.1 | **38.4** |
| | Linear Probing | **93.3** | 40.0 | 75.4 | 4.8 | 77.0 | 19.6 | 50.7 | 27.9 | 45.5 | 25.5 |
| | LP-FT | 93.1 | 38.2 | **82.5** | 5.1 | 77.6 | 23.1 | 52.8 | 30.8 | **67.1** | 37.4 |
| | LP-FT (WiSE) | **94.0** | 39.7 | 81.6 | 6.2 | 77.9 | 22.3 | **53.2** | 31.0 | 61.0 | 33.7 |
| | Reweight-E | 91.2 | 38.9 | 81.8 | 5.2 | 76.9 | 13.1 | 48 | 17.9 | 62.3 | **38.4** |
| | Reweight-Y×E | 91.3 | 39.0 | 81.5 | 5.3 | 72.2 | 17.4 | 41.6 | 30.0 | 57.5 | 37.6 |
| | CFA | 92.8 | 39.2 | **82.6** | 5.6 | **78.1** | 22.8 | 52.5 | 30.8 | **67.2** | **38.5** |
| | CFA (WiSE) | **93.1** | **40.4** | 79.6 | **6.4** | **78.2** | **23.8** | 52.6 | **33.4** | 59.8 | 34.3 |

**Postprocessing with WiSE-FT (Wortsman et al., 2022)**  After finetuning using the three baseline methods and our proposed CFA, we also apply WiSE-FT (Wortsman et al., 2022) with $\alpha = 0.5$ to postprocess the model, which just takes an average of initial and finetuned model parameters in the parameter space. It has been shown that Wise-FT can improve the model performance (especially OOD performance) in some cases (Wortsman et al., 2022). *Note:* For the CLIP experiments, we interpolate the finetuned model with the zero-shot CLIP encoder and classification head. For the DINOv2 experiments, since there is no zero-shot classification head available, we interpolate the finetuned models with the linear probing/stage-1 CFA results. Consequently, we do not perform WiSE-FT on full finetuning and reweighting baselines since they do have linear-probed heads.

**Implementation of CFA**  Empirically, we make small modifications to Stage 1 to divide it into two steps: i) Train $W_2$ on the domain labels with reweighting until it converges with $W_1$ fixed to its zero-shot weight; ii) then train $W_1$ on the class label with reweighting, and encourage the orthogonality between $W_1$ and the fixed $W_2$ with a $\ell_2$ regularization term, $\|W_1^\mathsf{T} W_2\|_F^2$. We also use reweighting for the class labels when performing the linear probing for LP-FT for a fair comparison. Following Sec. 2.1, we normalize the row vectors of $W_1$ and $W_2$ to the unit norm and normalize the outputs of $\Phi$ also to the unit norm. In addition, in the linear probing stage of CFA and LP-FT, we constrain $W_1$ and $W_2$ to a subspace determined by the zero-shot linear classifier of CLIP, as we find it can improve the final OOD performance. Besides, we find that in Stage 2, it is empirically sufficient to train the encoder with a very small $\lambda$ value, which is a loss coefficient in (3), and we deploy $\lambda = 0$ in Stage 2 to reduce compute cost. In both Stage 1 and Stage 2, we use the AdamW (Loshchilov & Hutter, 2017) optimizer with a cosine annealing scheduler (Loshchilov & Hutter, 2016). More details and hyperparameters can be found in Appendix C.

**Empirical Conclusions**  The results of our empirical experiments are presented in Table 1. Observing the results, we conclude that: **a)** Compared with full finetuning, LP-FT and reweighting, our CFA can improve the performance of pretrained models on OOD data for compositional generalization; **b)** WiSE-FT can further improve the OOD performance of all methods in most cases (when WiSE-FT fails, it will fail on both ID and OOD); **c)** While CFA enjoys a superior OOD performance, its in-distribution (ID) performance is maintained around the same level of full finetuning, which is a desired property. Also, we notice that,

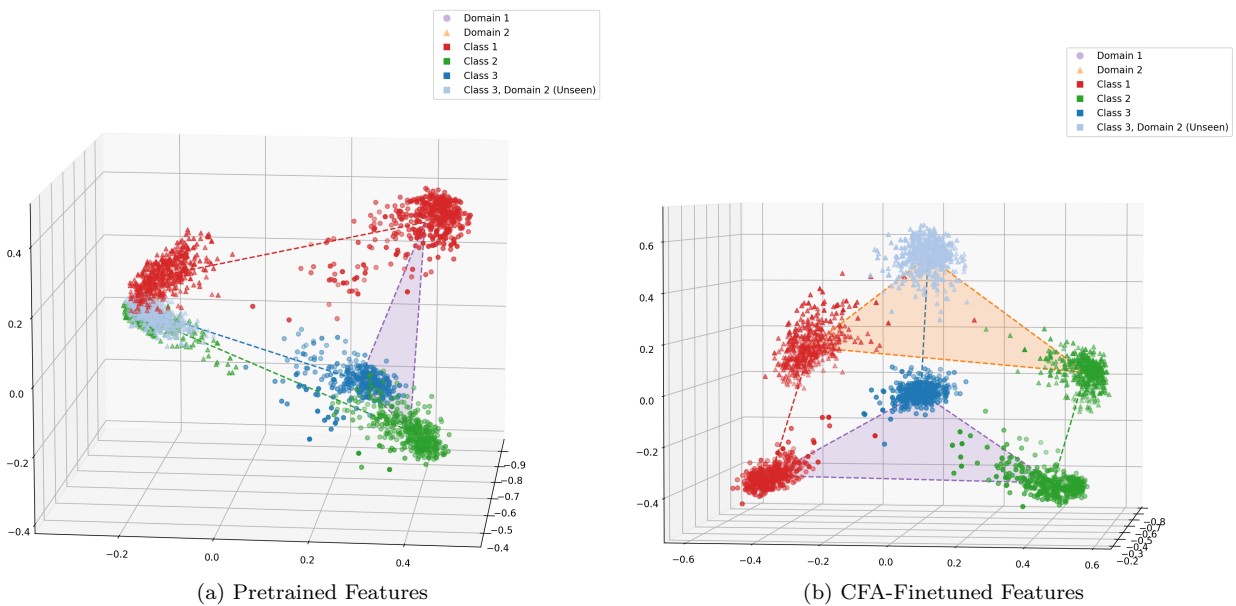

Figure 5: Visualization of the features for CLIP ViT-B/16 before and after the finetunning of CFA. **Left**: Feature extracted using pre-trained CLIP ViT-B/16 image encoder. **Right**: Feature extracted using CFA-finetuned CLIP ViT-B/16 image encoder.

although our CFA increases the performance of models on OOD data in CG tasks, there is still a gap between its ID and OOD performance, indicating that CG is quite a challenging task and needs future algorithm advancement to be further addressed.

### 3.3  Feature Visualization on Real-World Data

To empirically show that our two-stage algorithm promotes the encoder's ability to learn the compositional feature, we conduct a feature visualization study for the CLIP ViT-B/16 image encoder on the DomainNet (Peng et al., 2019) dataset. Specifically, we take the encoder both *before* and *after* finetuning with CFA, visualizing features for 2 domains and 3 classes. This resulted in 6 unique domain-class combinations (5 out of which are present in the training set). The visualization in Figure 5b clearly shows that the features finetuned with CFA conform to a compositional feature structure. In contrast, the features of the original CLIP model (Figure 5a) do not exhibit this structure. This visualization not only demonstrates the feature alignment ability of CFA but also provides further evidence of its effectiveness for large neural networks trained on real-world data.

### 3.4  Partial Availability of Domain Labels

In real-world applications, training samples may not be provided with domain labels, which may affect the applicability of CFA. We divide this problem into two scenarios: i) Domain labels are partially available; ii) Domain labels are completely unavailable.

Corresponding experiments on the Office-Home dataset are designed and conducted to demonstrate that the CFA remains effective even in the absence of domain labels. Under the first condition, we conduct experiments by only using 10%, 20%, and 50% of the domain labels for Stage-1 to learn the linear head for finetuning in Stage-2. We repeat each experiment 3 times using different subsets of the data and present the average result to mitigate the randomness caused by subsampling. For the second scenario, we propose that we can leverage the zeroshot ability of CLIP to predict the domain labels. To be specific, we first manually design a set of labels that can describe the domains of the data (for the experiment we just use the original domain names). Then, we perform the zeroshot classification on the training data and the domain labels using CLIP model. Finally, we take the predicted domain labels as ground-truth domain labels and use them in Stage-1. We adopt the hyperparameters shown in the manuscript and present the mean accuracy of CFA

Table 2: CFA with partial availability of domain labels. Experiments are conducted on Offce-Home.

| Method | Domain label ratio | ID | ID (WiSE) | OOD | OOD (WiSE) |
|---|---|---|---|---|---|
| CFA | 100% (Original) | 94 | 93.1 | 54.3 | 56.9 |
| | 50% | 94 | 93.3 | 53.8 | 56.9 |
| | 20% | 93.9 | 93.1 | 53.8 | 56.7 |
| | 10% | 94.0 | 93.2 | 53.4 | 55.9 |
| | 0% (CLIP Predict) | 94.1 | 93.0 | 52.0 | 53.6 |
| Finetune | 0% | 94.3 | 93.7 | 51 | 52.5 |
| LP-FT | 0% | 93.5 | 93.0 | 43.9 | 42.8 |

Table 3: Domain label prediction with partial training label availability. Experiments are conducted on Office-Home.

| Domain Label Ratio | 0% (Zero-Shot) | 10% | 20% | 50% | 100% |
|---|---|---|---|---|---|
| Avg. #Data per Domain | 0 | 308 | 616 | 1541 | 3082 |
| Accuracy | 61.5 | 82.0 | 84.6 | 85.5 | 86.3 |

and WiSE-FT over 3 seeds on the Office-Home dataset. From results in Table 2, we can see our CFA works well as domain labels are partially available.

The mild reliance of CFA's performance on domain label availability is beneficial for the practical application of this method but may raise questions about why the reliance is so mild. We believe there are two main reasons: i) the number of domains is small (4-6 domains for each 4 dataset of CG-Bench), so the available data per domain is relatively abundant, and ii) domain labels are easier to predict than class labels since image style or background information is highly indicative of the domain label, and these visual features are easy to capture by neural networks.

We clarify this by conducting a simple ablation study: in the Office-Home dataset with 4 domains, we fit linear classifiers to CLIP features for different amounts of domain labels (1%, 2%,..., 100% of the training data), and show the test prediction accuracy for domains below. Each experiment is repeated three times with different sampling seeds, and mean accuracy is reported in Table 3.

From the results, we can observe that as the domain labeling ratio increases from 10% to 100%, the domain prediction accuracy modestly improves from 82.0% to 86.3%, indicating a relatively small enhancement. This suggests that the domain label is indeed quite easy to predict, which could explain why our CFA can work well with a partial availability of domain labels. Furthermore, the zero-shot prediction accuracy of the CLIP ViT-B/16 model for domain labels is 61.5%, significantly higher than a random guess (25%). This outcome supports the notion that CFA with CLIP-predicted domain labels can also improve over vanilla finetuning.

**Class Label Availability** As for class labels, we do require their full availability for the training set, since our work focuses on *supervised finetuning* of pretrained encoders. Meanwhile, although CLIP is not explicitly supervised by domain and class labels, the texts in the image-text pairs (used for CLIP pretraining) provide abundant class and environment information. Furthermore, CLIP's unsatisfactory zero-shot performance on our CG-bench indicates the need for supervised finetuning to improve its effectiveness. On the other hand, DINOv2 is a self-supervised pretraining method, thus it needs to be supervised finetuned before applying to downstream classification tasks. Overall, the above statement wants to justify that the class label availability cannot be waived for the compositional generalization task considered in our paper.

**Ablation Studies** We conducted ablation studies on i) the choice of normalized head v.s. unconstrained head, ii) the choice of frozen head v.s. trainable head, and iii) the loss coefficient $\lambda$ in Stage 2 on CLIP model. Due to the page limit, we defer results and details to Appendix C. Also, we discuss the training stability, overhead and performance gain of CFA in Appendix C.

## 4 Related Works

**OOD Generalization** In OOD generalization, the model is trained on labeled data from a limited number of known domains, and the goal is to improve the performance of the models so that they can better generalize to previously unseen or new test domains (Blanchard et al., 2011). A common approach to tackle

OOD generalization is domain-invariant learning, which aims to learn a representation of data that has an invariant distribution over different training domains. Previous works taking this approach match domains in feature space either by aligning moments (Sun & Saenko, 2016) or using an adversarial loss (Ganin et al., 2016). However, these methods are later pointed out to be generally inadequate (Zhao, 2019). Another popular approach is to learn the optimal invariant predictors. Taking this approach, an invariant risk minimizer (IRM) optimizes a highly non-convex bi-level objective and simplifies the optimization using a penalty regularized objective. However, Rosenfeld et al. (2021); Kamath et al. (2021); Ahuja et al. (2021) theoretically show that these algorithms fail even in simple data models. Similarly, Wang et al. (2022) proposed Invariant-feature Subspace Recovery (ISR), which recovers the subspace spanned by the invariant features, and then fits predictors in this subspace. Distributionally robust optimization (Sagawa et al., 2020) is also a choice to tackle OOD generalization, which optimizes models over a worst-case distribution that is perturbed around the original distribution. In addition to the methods designed for training a model from scratch, recent works (Kumar et al., 2022; Wortsman et al., 2022; Goyal et al., 2022) also discuss increasing the OOD accuracy over the pretrained model. While these methods provide impressive empirical improvements on pretrained models, theoretical explanations are yet to be provided.

**Composition Generalization** In the computer vision literature, previous research has investigated attribute-object compositions (also referred to as compositional zero-shot learning) (Misra et al., 2017; Nagarajan & Grauman, 2018; Purushwalkam et al., 2019; Naeem et al., 2021; Nayak et al., 2023; Hao et al., 2023), with the goal of predicting attributes of an object in addition to its class. For instance, in a binary image classification task involving `cat` versus `tiger`, a classifier might be required to predict whether the animal is `old` or `young` alongside the conventional class prediction. In contrast, compositional generalization (CG) has a different focus. It concentrates purely on predicting the object class (e.g., `cat` versus `tiger`). Specifically, CG tasks aim to accurately identify `young cat` images as `cat`, even when the training data consisting only of `young tiger` and `old cat` instances and lacking `young cat` images. This scenario introduces an out-of-distribution (OOD) shift, and the challenge lies in developing models robust to such shifts. While compositional zero-shot learning (CZSL) can decouple attributes from objects, it is not universally adaptable for OOD generalization tasks. This approach relies on powerful vision-language models (VLMs) such as CLIP, while CG does limit the type of image classifiers. However, in certain real-world domains, such as remote sensing or medical imaging, there is a lack of paired image-text data to train strong VLMs. Therefore, adopting *self-supervised* models such as MAE (He et al., 2022) and DINO (Caron et al., 2021) presents a more practical strategy for these domains (Cong et al., 2022; Wanyan et al., 2023). As shown in Table 1, our CFA can work with both VLMs and self-supervised models. In contrast, CZSL can not be directly applied to self-supervised models. Besides, Sivaprasad et al. (2022) explores CG under a slightly simplified premise, where only one random domain is masked for each class. In addition, their method is built upon ResNet (He et al., 2016b) and does not scale well with modern transformer architectures.

## 5 Conclusion

This paper delves into the challenge of Compositional Generalization (CG) in machine learning, focusing on generalization to unseen domain-class combinations. By developing CG-Bench, a suite of benchmarks from real-world image datasets, we highlighted the shortcomings of prevalent pretraining-finetuning frameworks in tackling this challenge. Our proposed solution, the Compositional Feature Alignment (CFA), offers a promising approach to improve the CG performance of pretrained models, as evidenced in our experiments. Despite these advances, our study is not without limitations. Our experiments are currently limited to the base-sized ViT models, and our empirical studies draw from a restricted number of datasets of limited size. As we strive to overcome these limitations in future work, we look to include larger models and diversify our benchmark suite, exploring alternative data sources beyond images. We invite the broader machine learning community to join us in the ongoing exploration of the important challenge of CG.

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

# A    Benchmark Curation

We introduce the curation process of DomainNet (Peng et al., 2019) in Section 3.1. In this section, we provide more details of the proposed benchmark on the remaining datasets.

**OfficeHome (Venkateswara et al., 2017)** consists of 65 classes of objects depicted in four domains: {`Art`, `Clipart`, `Product Image`, `Real World Image`}. The ID and OOD splits are created in the same process as the DomainNet. The average top-1 accuracy is measured to evaluate the model performance on both sets.

**iWildCam (Beery et al., 2020)** is a dataset designed for classifying 182 classes of wild animals in camera traps. In the original iWildCam, each camera trap is defined to be a domain, resulting in 324 domains that are labeled as camera IDs. Such a definition of domains ignores the semantic similarities among them. For example, two camera traps may both be set in forests, however, the original iWildCam assigns them to different domains. Consequently, images taken in the forests are distributed in both ID and OOD sets regardless of the fact that they are from very similar environments. Observing such deficiencies in the original iWildCam, we are motivated to reassign the domain labels to the data. We propose 6 domains for the dataset: {`colored forest`, `gray-scale forest`, `colored trail`, `gray-scale trail`, `colored savanna`, `gray-scale savanna`}. To assign the new domains to the images, we take advantage of the zero-shot classification ability of the CLIP models. Specifically, we design 10 prompts for each domain and perform the zero-shot prediction of the domains for each image using the assembled prompts. New domain labels are generated for both training and test sets, resulting in a dataset that is consistent with our CG setting: the $E \times K$ matrix is sparse, and the test set contains domain-class pairs that have few or no samples in the training set. We take the domain-class pairs with less or equal to 20 samples in the training set as the OOD samples in the test set. In addition to measuring the average top-1 accuracy, we also measure the F1-macro score for both the ID and OOD sets.

**FMoW (Christie et al., 2018)** is a dataset that aims to classify utility types from satellite images. This dataset has two sets of domains: *years* and *regions*. The ID and OOD of the dataset are split according to the *years*, i.e., all the satellite images in the test set taken before 2016 are ID samples, and images after 2016 are OOD samples. In our CG-Bench, we only perform evaluations on *region* domains. To be specific, the dataset has $K = 62$ classes and $E = 5$ domains: {`Asia`, `Europe`, `America`, `Africa`, `Oceania`}. The $E \times K$ matrix is naturally sparse because certain utilities were only introduced post-2016 in some underdeveloped regions, leading to the creation of OOD entries. Therefore, FMOW is readily suited for our CG setting without the need for additional curation. We measure the average top-1 accuracy for the ID set and the worst region accuracy for the OOD set as suggested by previous works.

# B    Proof of Theorem 1

We first restate the setting, and introduce some notations and definitions. Then we present two lemmas useful for the proof of our main theorem. Finally, we restate our main theorem (Theorem 1) and provide a proof.

**Notation and Setting**    We denote $\mathbf{Z} = [\phi(x_1), \ldots, \phi(x_n)] \in \mathbb{R}^{d \times N}$, $\mathbf{Y} = [y_1, \ldots, y_N]$, and $\mathbf{E} = [e_1, \ldots, e_N]$ as the stack of features, class labels, and environment labels, respectively. In the context of the unconstrained feature model, the optimization objective of Stage 2 is transformed to:

$$\min_{\mathbf{Z}} \frac{1}{KN} \ell_{\text{CE}}(\beta_1 \cdot W_1 \mathbf{Z}, \mathbf{Y}) + \lambda \frac{1}{EN} \ell_{\text{CE}}(\beta_2 \cdot W_2 \mathbf{Z}, \mathbf{E}) \quad \text{s.t.} \quad \mathbf{Z} \in \mathcal{U}(d)^N \tag{6}$$

Furthermore, we use $W^{(j)}$ to refer to the $j$-th row vector of matrix $W$, and $z_i$ represents the $i$-th column vector of $\mathbf{Z}$, which is also the learned feature of the $i$-th training sample.

Below, we define simplex-encoding label (SEL) matrices, which are introduced in Thrampoulidis et al. (2022), and also consider SVD of the two heads, $W_1$ and $W_2$.

**Definition 2** (Simplex-Encoding Label Matrices)**.** $\mathbf{S}_1 \in \mathbb{R}^{K \times N}$ *and* $\mathbf{S}_2 \in \mathbb{R}^{E \times N}$ *are simplex-encoding label (SEL) matrices for classes and domains, respectively, such that*

$$\forall c \in [K], i \in [N]: \quad \mathbf{S}_1[c,i] = \begin{cases} 1 - 1/K & , \quad c = y_i \\ -1/K & , \quad c \neq y_i \end{cases} \tag{7}$$

$$\forall c \in [E], i \in [N]: \quad \mathbf{S}_2[c,i] = \begin{cases} 1 - 1/E & , \quad c = e_i \\ -1/E & , \quad c \neq e_i \end{cases} \tag{8}$$

*In other words, the $i$-th column of $\mathbf{S}_1$ is a $K$-dimensional vector, $(\boldsymbol{e}_{y_i} - \frac{1}{K}\mathbf{1})$, where $\boldsymbol{e}_j$ represents the $i$-th standard basis vector. Similarly, the $i$-th column of $\mathbf{S}_2$ is a $E$-dimensional vector, $(\boldsymbol{e}_{e_i} - \frac{1}{E}\mathbf{1})$.*

**Definition 3** (SVD of Heads)**.** *For $W_1$ and $W_2$, we consider their compact SVD as*

$$W_1 = U_1 \Lambda_1 V_1^{\mathsf{T}} \quad and \quad W_2 = U_2 \Lambda_2 V_2^{\mathsf{T}} \tag{9}$$

*Specifically, $\Lambda_1, \Lambda_2$ are positive diagonal matrices and $U_1, V_1, U_2, V_2$ have orthonormal columns.*

Now, we focus on the first term of (6), and study its optimum with theoretical tools from Thrampoulidis et al. (2022).

**Lemma 1** (Optimum of Class Loss)**.** *Assuming the feature dimension $d$ is at least $K$, and training data exists for each class, and $W_1$ is normalized such that $W_1 \in \mathcal{U}(d)^K$. Additionally, we assume $\beta_1$ is sufficiently large. Then, for any constant norm magnitude $a > 0$, the global minimum of the following objective,*

$$\min_{\mathbf{Z}} \frac{1}{KN} \ell_{\mathrm{CE}}(\beta_1 \cdot W_1 \mathbf{Z}, \mathbf{Y}) \quad s.t. \ \|z_i\|_2 = a , \tag{10}$$

*satisfies*

$$\mathbf{Z}^* = \gamma_1 V_1 \Lambda_1^{-1} U_1^{\mathsf{T}} \mathbf{S_1} , \tag{11}$$

*where $\gamma_1 = \frac{a}{1-1/K}$.*

*Proof.* From Theorem 1 of Thrampoulidis et al. (2022), we know that the optimum of (10) shall satisfy

$$W_1 \mathbf{Z}^* = \gamma \mathbf{S}_1 \tag{12}$$

where $\gamma > 0$ is a scaling factor and $\mathbf{S}_1$ is defined in Definition 2.

With (12) and the norm constraint $\|z_i\|_2 = a$, it is obvious that $\mathbf{Z}^*$ should span the same feature subspace as $W_1$, since the cross-entropy loss is monotone such that a larger magnitude of $W_1 \mathbf{Z}^*$ leads to smaller loss. Therefore, we can express $Z^*$ as

$$\mathbf{Z}^* = \gamma_1 V_1 \Lambda_1^{-1} U_1^{\mathsf{T}} \mathbf{S_1} , \tag{13}$$

and the scaling factor can be easily determined by considering $\|W_1^{(i)}\|_2 = 1$ and $\|z_i\|_2 = a$.

$\square$

We can study the optimum of the second term of (6) in the same fashion, and obtain similar results.

**Lemma 2** (Optimum of Domain Loss)**.** *Assuming the feature dimension $d$ is at least $E$, and training data exists for each domain, and $W_2$ is normalized such that $W_2 \in \mathcal{U}(d)^E$. Additionally, we assume $\beta_2$ is sufficiently large. Then, for any constant norm magnitude $b > 0$, the global minimum of the following objective,*

$$\min_{\mathbf{Z}} \frac{1}{EN} \ell_{\mathrm{CE}}(\beta_2 \cdot W_2 \mathbf{Z}, \mathbf{E}) \quad s.t. \ \|z_i\|_2 = b , \tag{14}$$

*satisfies*

$$\mathbf{Z}^* = \gamma_2 V_2 \Lambda_2^{-1} U_2^{\mathsf{T}} \mathbf{S_2} , \tag{15}$$

*where $\gamma_2 = \frac{b}{1-1/E}$.*

*Proof.* The proof is the same as that of Lemma 1. □

With Lemma 1 and Lemma 2, we can prove that the learned feature vector of each training sample can be decomposed as a linear combination of two vectors depending on the class label and domain label, respectively, while the two vectors live in orthogonal feature subspaces. This indicates that the learned features conform to a compositional feature structure satisfying Definition 1.

Notice that Theorem 2 (the restatement of Theorem 1) is slightly different from Theorem 1. This is because we found a small issue in the original proof for Theorem 1. We fixed the issue before the supplementary material submission deadline, which led to a slightly different expression of $z_i^*$. However, the same main conclusion still applies to the updated theorem (Theorem 2): the learned features comply with the compositional feature structure in Definition 1, which justifies the algorithm design of our algorithm design. We will revise Theorem 1 in our main text correspondingly in the revision.

**Theorem 2** (Feature Alignment (Theorem 1 Restated)). *Assuming the feature dimension d is no smaller than $K + E$, and training data exists for each class and domain (though not necessarily for each domain-class combination), and $W_1$ and $W_2$ are normalized and span orthogonal subspaces such that $W_1 \in \mathcal{U}(d)^K, W_2 \in \mathcal{U}(d)^E$ and $W_1 W_2^\mathsf{T} = \mathbf{0}$. Additionally, we assume $\beta_1, \beta_2$ are sufficiently large. The global minimum of (6) results in the following: for any $i \in [N]$, denote $z_i$ as the i-th column vector of $\mathbf{Z}$, we have*

$$z_i^* = W_1^\mathsf{T} \boldsymbol{a}_{y_i} + W_2^\mathsf{T} \boldsymbol{b}_{e_i} \tag{16}$$

*where $\boldsymbol{a}_{y_i} \in \mathbb{R}^K$ is a vector depending on the class label $y_i$, and $\boldsymbol{b}_{e_i} \in \mathbb{R}^E$ is a vector depending on the domain label $e_i$.*

*Proof.* Lemma 1 and Lemma 2 derive the optimum of the two loss terms of (6), respectively. For the linear combination of the two loss terms with coefficient $\lambda$, it is straightforward to see that the optimum $Z^*$ should also be a linear combination of (11) and (15), since these two optima stay in orthogonal subspaces spanned by $W_1$ and $W_2$, respectively (Stage 1 ensures that $W_1$ and $W_2$ live in orthogonal subspaces). In other words, the first loss term $\frac{1}{KN}\ell_{\mathrm{CE}}(\beta_1 \cdot W_1\mathbf{Z}, \mathbf{Y})$ does not affect the converged direction of $\mathbf{Proj}_{W_2}Z^*$, i.e., $Z^*$ projected into the subspace spanned by $W_2$, and vice versa. However, the first term affects the magnitude of $\mathbf{Proj}_{W_2}Z^*$ since we have the norm constraint $\mathbf{Z} \in \mathcal{U}(d)^N$ in (6). Specifically, the larger $\lambda$ in (6) leads to larger magnitude of $\|\mathbf{Proj}_{W_2}Z^*\|_2$. Hence, we can express $Z^*$ as

$$\mathbf{Z}^* = a\gamma_1 V_1\Lambda_1^{-1}U_1^\mathsf{T}\mathbf{S_1} + b\gamma_2 V_2\Lambda_2^{-1}U_2^\mathsf{T}\mathbf{S_2} \tag{17}$$

where $a, b > 0$ are scaling factors (depending on $\lambda$) that ensure $Z^* \in \mathcal{U}(d)^N$.

By plugging in Definition 2, we can express the $i$-th column vector of $Z^*$, the learned feature of the $i$-th sample, as

$$z_i^* = a\gamma_1 V_1\Lambda_1^{-1}U_1^\mathsf{T}(\boldsymbol{e}_{y_i} - \frac{1}{K}\mathbf{1}) + b\gamma_2 V_2\Lambda_2^{-1}U_2^\mathsf{T}(\boldsymbol{e}_{e_i} - \frac{1}{E}\mathbf{1}) \tag{18}$$

Clearly, the first term of (18) depends on the class label $y_i$ only and the second term relies only on the domain label $e_i$. Besides, the two terms live in the feature space spanned by $V_1$ and $V_2$, respectively, which are determined by $W_1$ and $W_2$. Hence, we can be re-expressed (18) as

$$z_i^* = W_1^\mathsf{T} \boldsymbol{a}_{y_i} + W_2^\mathsf{T} \boldsymbol{b}_{e_i} \tag{19}$$

where $\boldsymbol{a}_{y_i} = a\gamma_1 U_1\Lambda_1^2 U_1^\mathsf{T}(\boldsymbol{e}_{y_i} - \frac{1}{K}\mathbf{1})$ and $\boldsymbol{b}_{e_i} = b\gamma_2 U_2\Lambda_2^2 U_2^\mathsf{T}(\boldsymbol{e}_{e_i} - \frac{1}{E}\mathbf{1})$. □

## C Details of Experiment

Here, we supplement Section 3 with more details about our experimental studies.

**Computational Expenses**     The experiments described in this paper are executed on NVIDIA RTX A6000 GPUs with 48GB memory, utilizing a total of 12 GPUs. The individual training runs for Office-Home, DomainNet, iWildCam, and FMoW require 0.2, 0.8, 0.5, and 0.2 GPU hours respectively. Each experiment is repeated ten times with varying random seeds, which alter the order of batch shuffling in the training loader.

**Hyperparameter Choices**     We present the hyperparameter settings for our CFA models in Table 4. The parameters for each stage are chosen based on model performances on the OOD validation set. Note that $\lambda$ is the domain loss coefficient in (3), and $\lambda_{\text{ortho}}$ is the coefficient for the orthogonality regularization loss $\|W_1^\mathsf{T} W_2\|_F^2$ that we use to ensure orthogonality of heads in Stage 1. The hyper-parameters in Stage 2 are also used for the two baseline algorithms, Full finetuning (FT) and LP-FT.

**Ablation Studies**     We conduct ablation studies on i) the choice of normalized head v.s. unconstrained head, ii) the choice of frozen head v.s. trainable head, and iii) the loss coefficient $\lambda$ in (3). The results are discussed as follows:

- *Head Normalization.*  In our implementation, we normalize the weight of our classification heads that has no bias terms, following recent works (Goyal et al., 2022; Wang et al., 2023) which show that head normalization is useful for CLIP finetuning. In this ablation study, we repeat the vanilla full finetuning experiment using unconstrained classification heads (i.e., with bias and without normalization). The test accuracies(%) are present in Table 5. From the results, we can see that the models finetuned with constrained heads have slightly better OOD performances compared to the unconstrained ones, justifying the use of the head normalization technique.
- *Trainable Heads v.s. Frozen Heads.*  In Stage 2, we finetune the models with *frozen* heads, $W_1$ and $W_2$, which are linearly probed in Stage 1. In this ablation study, we perform the full finetuning during Stage 2, i.e., training all the layers including the classification heads. The test accuracies(%) are shown in Table 6. The results indicate that using trainable heads or using frozen heads does not vary the finetuned model performance much. Therefore, this ablation study justifies that our algorithmic design of frozen classification heads is reasonable.
- *Domain Loss Coefficient.*  In our empirical implementation, we consider $\lambda = 0$ in (3) and only finetune with respect to $W_1$ in Stage 2, to reduce the training cost. The motivation for doing so is the observation that the coefficient $\lambda$ in Stage 2 does not affect the final performance. To justify this, we conduct the following ablation study: we run Stage 2 with the original two-term loss, (3), while adjusting the coefficient $\lambda$ for the domain prediction loss. The test accuracies(%) are shown in Figure 6. From the figure, one can observe that the value of $\lambda$ can be set to zero in Stage 2 without negatively affecting the final performance. Therefore, it is sufficient to consider $\lambda = 0$ in Stage 2 and only fine-tune the model with the first loss term in (3).

## C.1   Discussion on Training Stability and Overhead

Here we conduct an additional study to examine the training stability of our two-stage method, CFA. All of the following experiments are conducted for CLIP ViT-B/16 on the Office-Home dataset and are an average over 3 seeds.

Table 4: Hyperparameters for our algorithm.

| Model | CLIP | | | | | DINOv2 | | | | |
|---|---|---|---|---|---|---|---|---|---|---|
| Dataset | Stage 1 (Linear Probing) | | | Stage 2 (Fine-Tuning) | | Stage 1 (Linear Probing) | | | Stage 2 (Fine-Tuning) | |
| | $\lambda$ | $\lambda_{\text{ortho}}$ | Epochs | Epochs | Learning Rate | $\lambda$ | $\lambda_{\text{ortho}}$ | Epochs | Epochs | Learning Rate |
| OfficeHome | 1 | 100 | 200 | 3 | $10^{-5}$ | 1 | 100 | 200 | 3 | $5 \times 10^{-5}$ |
| DomainNet | 1 | 10000 | 200 | 3 | $10^{-5}$ | 1000 | 10 | 200 | 10 | $5 \times 10^{-5}$ |
| iWildCam | 10 | 10 | 200 | 5 | $10^{-5}$ | 1 | 100 | 200 | 5 | $10^{-5}$ |
| FMoW | 10 | 100 | 200 | 3 | $10^{-5}$ | 100 | 1000 | 200 | 4 | $5 \times 10^{-5}$ |

Table 5: The ablation study on the constraints on classification heads. Blue cells represent the normalized classification head with no bias; Orange cells indicate the heads are unconstrained.

| Methods | OfficeHome | | DomainNet | | iWildCam | | FMoW | |
|---|---|---|---|---|---|---|---|---|
| | ID | OOD | ID | OOD | ID | OOD | ID | OOD |
| FT | 94.3 | 51.0 | 82.0 | 7.5 | 74.5 | 16.5 | 65.8 | 38.7 |
| FT-Wise | 93.7 | 52.5 | 76.4 | 8.7 | 67.0 | 13.7 | 49.5 | 40.6 |
| FT | 94.3 | 50.8 | 82.0 | 7.5 | 74.9 | 16.2 | 65.8 | 38.5 |
| FT-Wise | 93.7 | 52.3 | 76.5 | 8.7 | 66.8 | 13.5 | 49.5 | 40.1 |

Table 6: The ablation study on the frozen classification heads v.s. trainable heads in Stage 2. Blue cells represent the fronzen heads; Orange cells indicate the heads are trainable.

| Methods | OfficeHome | | DomainNet | | iWildCam | | FMoW | |
|---|---|---|---|---|---|---|---|---|
| | ID | OOD | ID | OOD | ID | OOD | ID | OOD |
| Zeroshot | 89.2 | 50.3 | 61.7 | 6.6 | 13.7 | 6.9 | 20.4 | 18.8 |
| FT | 94.3 | 51.0 | 82.0 | 7.5 | 74.5 | 16.5 | 65.8 | 38.7 |
| FT (Wise) | 93.7 | 52.5 | 76.4 | 8.7 | 67.0 | 13.7 | 49.5 | 40.6 |
| FT | 94.3 | 51.4 | 82.1 | 7.3 | 75.3 | 16.4 | 65.8 | 38.5 |
| FT (WiSE) | 93.7 | 52.9 | 76.6 | 8.4 | 67.7 | 13.6 | 49.7 | 39.7 |
| LP-FT | 93.5 | 43.9 | 81.5 | 5.3 | 74.0 | 17.0 | 65.9 | 40.2 |
| LP-FT (WiSE) | 93.0 | 42.8 | 79.4 | 5.3 | 74.4 | 18.2 | 56.6 | 36.3 |
| LP-FT | 93.5 | 43.9 | 81.6 | 5.3 | 74.0 | 17.2 | 65.5 | 39.9 |
| LP-FT (WiSE) | 93.1 | 42.9 | 79.4 | 5.3 | 74.4 | 18.3 | 58.8 | 36.4 |
| CFA (LP-FT) | 93.7 | 53.2 | 81.6 | 7.3 | 74.0 | 18.3 | 65.3 | 41.6 |
| CFA (WiSE) | 93.1 | 56.4 | 76.5 | 9.2 | 74.6 | 19.7 | 53.5 | 36.6 |
| CFA (LP-FT) | 93.7 | 52.8 | 81.7 | 7.0 | 73.9 | 18.4 | 65.5 | 41.0 |
| CFA (WiSE) | 93.0 | 56.2 | 76.6 | 8.9 | 74.5 | 19.6 | 53.7 | 36.0 |

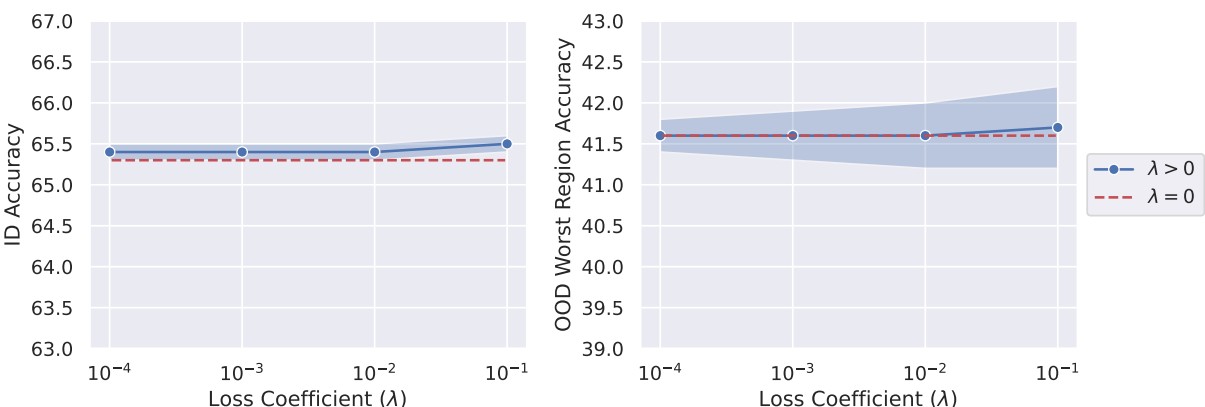

Figure 6: The ablation study on domain loss coefficients. The experiments are conducted on the FMoW dataset as an example. The red dashed lines are the performance of the model without supervision on the domain labels.

We first show that our method is stable to the number of iterations in Stage-1. From Table 7, we can see that as the linear heads are trained longer in Stage-1, the ID (WiSE) performance slightly increases and the OOD accuracy reaches the peak at 4000 to 6000 iterations. Except for the trade-off between ID and OOD performance, our method is stable with no drastic drop in accuracy. The 6000 iterations of Stage-1 was the hyperparameter used for experiments in Table 1.

We then demonstrate how training epochs in Stage-2 will affect the final performance of our model. Table 8 shows that for longer training epochs, the ID accuracy increases and the OOD accuracy decreases. Hence, in our paper, we used 3 epochs for the best OOD performance.

Table 7: Study on the training stability of CFA's Stage-1.

| LP Iterations | ID | ID (WiSE) | OOD | OOD (WiSE) |
|---|---|---|---|---|
| 2000 | **94.0** | 92.8 | 54.3 | 56.9 |
| 4000 | **94.0** | 93.1 | **54.3** | **57.3** |
| 6000 (Chosen) | **94.0** | 93.1 | **54.3** | 56.9 |
| 8000 | **94.0** | 93.6 | 53.3 | 56.5 |
| 10000 | **94.0** | **93.7** | 52.1 | 55.0 |

Table 8: Study on the training stability of CFA's Stage-2.

| Epochs | ID Acc | ID Acc (WiSE) | OOD Acc | OOD Acc (WiSE) |
|---|---|---|---|---|
| 3 (Chosen) | 94.0 | 93.1 | **54.3** | **56.9** |
| 5 | **94.1** | 93.5 | 52.3 | 56.3 |
| 10 | **94.1** | **93.7** | 52.2 | 55.7 |

**Parameter Overhead**   Compared with the vanilla full fine-tuning, our CFA introduces an additional linear classifier for domain labels in Stage-1. In terms of model parameters, we provide the parameter count of CLIP and DINOv2 on DomainNet, in Table 9. One can see that the additional parameters ($e$-classifier) introduced by CFA is negligible compared with the total number of parameters.

Table 9: Parameter count for CLIP and DINOv2 on DomainNet. $y$-classifier is the linear classifier for class labels, and $e$-classifier is the linear classifier for domain labels. Vision encoders, language encoders and $y$-classifier are modules shared between the vanilla fine-tuning and CFA, and the last column is additional parameters introduced by CFA.

| Parameters | Vision Encoder | Language Encoder | $y$-classifier | $e$-classifier (CFA only) |
|---|---|---|---|---|
| CLIP ViT-B | $86 \times 10^6$ | $91 \times 10^6$ | $177 \times 10^3$ | $3.1 \times 10^3$ |
| DINOv2 ViT-B | $86 \times 10^6$ | - | $265 \times 10^3$ | $4.6 \times 10^3$ |

**Compute Overhead**   The training cost of Stage-1 (linear probing) is much smaller compared to Stage-2 (backbone finetuning). Stage-1 requires only a single forward pass over training samples for feature gathering (features are stored in CPU memory), and then the linear probing can be achieved fastly on CPU. The training cost for Stage-2 of CFA is slightly less than standard full finetuning, as the last layer remains frozen during Stage-2 of CFA. Overall, we conclude that the additional training costs of CFA are mild (mostly coming from the forward pass over training samples), which is less than that of *one fine-tuning epoch*. In other words, compared with vanilla fine-tuning for $N$ epochs, our CFA's training compute cost is less than fine-tuning for $N + 1$ epochs. The modest compute overhead of CFA making it a practical method for addressing compositional generalization problems in real-world applications.

**Memory and Speed Overhead**   For CFA, the memory requirement of Stage-1 is less than Stage-2 since Stage-1 only involves forward passes of the image encoder (no backward pass). For Stage-2 of CFA, the memory usage and training speed are the *same as that of the vanilla finetuning*, since Stage-2 of CFA does not introduce additional parameters or loss for training.

### C.2   Discussion of the Performance Gain

From Table 1, one may think the performance gain of CFA over reweighting, a strong baseline method, is not significant. However, our main contribution lies in designing a principled algorithm for compositional generalization, a novel challenge in OOD generalization. As our theoretical analysis and feature visualization in Fig. 5 demonstrate (on Color MNIST and DomainNet, respectively), our method aligns features with a compositional structure suitable for compositional generalization. In OOD generalization research, it's typical for specialized algorithms to only modestly outperform baselines like ERM, as evidenced by extensive benchmarking in studies like DomainBed (Gulrajani & Lopez-Paz, 2021) and WILDS (Koh et al., 2021). Thus, we consider CFA's performance gain to be meaningful and convincing. Additionally, the mild implementation complexity and training costs (discussed in Sec. C.1) make CFA a viable method for enhancing compositional generalization.

Moreover, we want to emphasize that real-world datasets used in our paper have lots of labeling issues (for both class and domain labels), which may prevent CFA from obtaining greater performance gain. Notably,

CFA employs both class and domain labels for training, leading to an increased susceptibility to label noise, particularly from domain labels. Below, we illustrate two types of labeling noise encountered:

- *Mislabeling*: We observe that the DomainNet dataset contains many mislabeled images. Many images in the "Clipart" domain are actually real photos of a single object on a white background, which should be categorized into the "Product" domain. On the other hand, numerous class labels are also mislabeled: images in the "bush (plant)" class contain pictures of President George Bush of the USA; the "cooler" class includes electric fan images, which are incorrectly categorized; the "square (shape)" class contains table, which should be placed into the "table" class instead. Note: We only manually examined a very tiny subset of the DomainNet dataset (Gulrajani & Lopez-Paz, 2021), which comprises 0.6 million images, and have already found many mislabeled images. Therefore, overall, we believe the mislabeling ratio is not negligible.

- *Ambiguous Labels*: We observe that certain domains contain a large number of images that are visually similar to those in another domain. For example, in both Office-Home and DomainNet, the images in the "Product" domain are real photos of objects, making them almost indistinguishable from their counterparts in the "Real" domain. The distinguishing feature of the "Product" domain is that its images all have a white background; however, some images in the "Real" domain also share this characteristic. Additionally, in DomainNet, the "Infograph" domain contains many images stylistically similar to those in "Clipart" or "Real"; some images in the "Painting" domain are sketches, despite the presence of a separate "Sketch" domain. This ambiguity issue extends to class labels as well. In DomainNet, the classes "cup," "coffee cup," and "mug" lack clear stylistic distinctions.

In addition to these issues, other aspects of these datasets may affect learning. For instance, the iWildCam dataset includes a class labeled "empty", signifying the absence of animals, which comprises a significant portion of the dataset.

### C.3 Effect of the Orthogonality Regularization

To better understand the impact of the orthogonality regularization on the optimization process, we conducted an ablation study on Stage-1 of our CFA method using the DomainNet dataset. In this study, we optimized the objective in Equation 1 with different orthogonal regularization coefficients ($\lambda_{\text{ortho}}$) for 20 epochs. The results of this ablation study are visualized in Fig. 7.

As shown in Fig. 7, the orthogonality loss (measured by $\|W_1^T W_2\|_F$) decreases as the linear probing epochs progress, indicating that the orthogonality regularization effectively enforces the orthogonality constraint between the two classifier heads $W_1$ and $W_2$. The rate of decrease in orthogonality loss is proportional to the value of the regularization coefficient $\lambda_{\text{ortho}}$, with larger values resulting in faster convergence towards orthogonal classifier heads.

This ablation study demonstrates that the orthogonality regularization term in our CFA method is crucial for obtaining orthogonal classifier heads $W_1$ and $W_2$, which are subsequently used in Stage-2 of our approach. By effectively enforcing the orthogonality constraint, our method ensures that the learned features are well-suited for compositional generalization tasks.

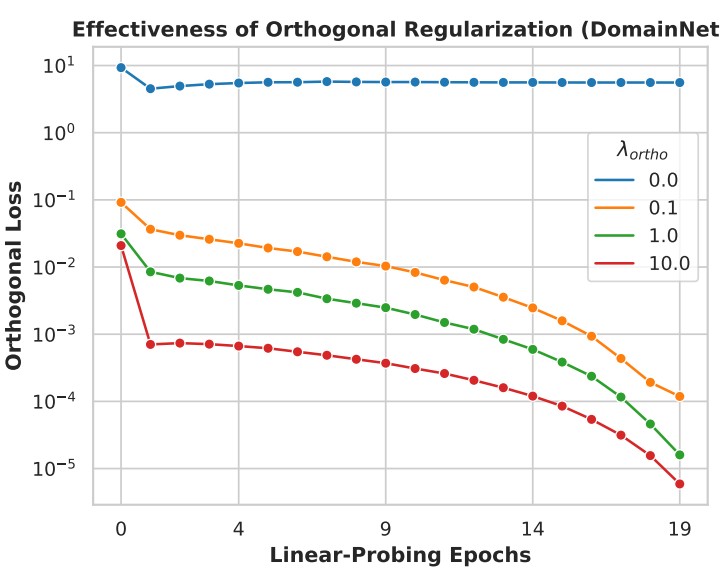

Figure 7: Effectiveness of the orthogonality regularization during Stage-1 linear probing on DomainNet. The orthogonality loss, $|W_1^T W_2|F$, is plotted against the number of epochs for different values of $\lambda ortho$. Larger $\lambda_{ortho}$ leads to faster convergence towards orthogonal $W_1$ and $W_2$.

