# OpenReview forum: "Enhancing Compositional Generalization via Compositional Feature Alignment"
_TMLR — Accepted by TMLR_

### Review · Reviewer_kFXZ · 2024-03-05

**Summary Of Contributions:**

The paper proposed a, two-stage, fine-tuning approach for achieving compositional generalization between class labels & domains.
The approach involves : 1-multi-head training on classes & domains, 2-finetuning the features through the heads. It also makes use for weight averaging (WiSE-FT) to improve the performance further. Results are presented on a combined dataset consisting of Office-home, DomainNet, and iWildCam and WILDS datasets.

**Audience:**

Yes

**Claims And Evidence:**

No

**Requested Changes:**

-The legend in Figure 2 is broken (some shapes should be triangles instead of squares), axes are not labeled -- do the axes represent various features?
-Definition 1 seems arbitrary and overly constrained. It seems tailored to the datasets under study, this definition does not seem suitable for realistic data.

**Strengths And Weaknesses:**

Strengths:
1-the presented method seems relatively simple and therefore reproducible.

Weaknesses:
1-The main weakness seems to be that the method is specifically tailored to the dataset and may have limited impact on real-world problems. Specifically there seems to be two key assumptions: (i) - domain labels are available, and (ii) - the domain labels and class labels are mutually independent. The second assumption is implicit in the algorithmic constraint $W_1 W_2^T = 0$. Yes, section 3.4 has a discussion on having limited domain data labels, nevertheless the fact that the data generative process cleanly consist of just a few domains is already a limiting factor. For many real world applications (ii) is not a valid assumption, e.g. imagine different species being recorded with different instruments, in this case the domains are not independent with the classes so it would seem a key design decision is violated.
2-As the authors mention in the paper, the evaluations were conducted with "based-size" (i.e. small) ViT models. Therefore, the relevance of this algorithm at large scales is dubious.

---

> ### Author Response · Authors · 2024-03-28
> **Author Response to Reviewer kFXZ**
>
> ### **Weakness 1 (Key Assumptions)**
>
> For assumption (i), we have conducted experiments in Sec 3.4 to show that our method can work in the case of partially available domain labels or even no available domain labels (in this case, we can use CLIP to zero-shot predict domain labels). For assumption (ii), it seems you misunderstand it as **statistical independence** between domain labels and class labels; however, we want to clarify that the assumption is actually **causal independence** instead (technically speaking, we assume class-related features are orthogonal to domain-related features on the **causal** level). The two notions are quite different. It is quite common that the domain label $e$ is causally independent of $y$ but statistically dependent, where this statistical dependence is also called spurious correlation. For instance, if we use your example of different species (classes) being recorded with different instruments (domains), in this case, the instruments are causally independent of the species, since the species of the animals being captured are determined by their DNA instead of the instrument. However, statistically, there might be a correlation between instruments and species, and this is exactly the **spurious correlation** that our research is trying to overcome. If one just uses ERM, the model exploits the spurious correlation easily; if one tries to remove the statistical dependence, that is reweighting, but this turns out not to be practically useful enough to tackle the spurious correlation problem [1]. Our motivation is to mitigate the learning of spurious correlations between domain-related features and class-related features with a special feature learning algorithm.
>
> ### **Weaknesses 2 (Model Size)**
>
> Unfortunately, our computing resources do not allow extensive experiments with ViT-L. Moreover, the literature on OOD robustness mostly uses ViT-B. Even for papers using both ViT-B and ViT-L, they do not observe inconsistencies in OOD robustness as the model size scales up [2][3][4]. So conclusions based on ViT-B are generally believed by the OOD research community to be generalizable to larger sizes as well.
>
> ### **Requested Change for Figure 2**
>
> In Figure 2, we used different colors to represent samples from different domains and different shapes to represent different classes. Therefore, the squares in our legend indicate the colors of the domains, and the legend is designed in this way. The three axes of the figure represent the three dimensions of feature space. There is no explicit meaning for each basis and consequently, we did not mark the axes.
>
> ### **Requested Change for Definition 1**
>
> As stated above, our key assumption is the causal independence between classes and domains, and we materialize this notion by considering that the class-related features and domain-related features are orthogonal to each other. The orthogonality is important, as linear classifiers that only rely on the class-related features will not utilize the domain-related features. The Gaussian modeling of features in Definition 1 is also common in the literature [5][6]. Besides, as we prove in Theorem 1, the global optimum of our CFA objective has feature collapse, which indicates that the covariance terms ($\Sigma_1^y$ and $\Sigma_2^e$) will have vanishing magnitudes. In practice, the absolute feature collapse only happens after a very long time of optimization, as demonstrated by neural collapse research [7][8]. With finite-time optimization, one usually observes relative feature collapse as features are Gaussian-like (centered around a centroid), which matches the Gaussian feature modeling in our Definition 1.
>
> # References
>
> [1] Sagawa et al. Distributionally Robust Neural Networks for Group Shifts: On the Importance of Regularization for Worst-Case Generalization. ICLR 2020
>
> [2] Wortsman et al. Robust fine-tuning of zero-shot models. CVPR 2022
>
> [3] Goyal et al. Finetune like you pretrain: Improved finetuning of zero-shot vision models. CVPR 2023
>
> [4] Gadre et al. DataComp: In search of the next generation of multimodal datasets. NeurIPS 2023
> [5] Wang et al. Provable Domain Generalization via Invariant-Feature Subspace Recovery. ICML 2022
>
> [6] Chen et al. Iterative Feature Matching: Toward Provable Domain Generalization with Logarithmic Environments. NeurIPS 2022
>
> [7] Papyan et al. Prevalence of neural collapse during the terminal phase of deep learning training. PNAS 2020
>
> [8] Kothapalli. Neural Collapse: A Review on Modelling Principles and Generalization. TMLR 2023

---

### Review · Reviewer_kZdn · 2024-03-12

**Summary Of Contributions:**

This paper introduces a novel approach to tackle the a new challenge in machine learning named as compositional generalization (CG), which aims at enhancing generalization to unseen domain-class combinations. The study proposes a method called Compositional Feature Alignment (CFA) to boost the CG performance of pretrained models via two-stage training. It also contributes to the field by developing CG-Bench, a benchmark suite derived from real-world image datasets, to assess the effectiveness of the proposed approach.

**Audience:**

Yes

**Claims And Evidence:**

Yes

**Requested Changes:**

The paper introduces a promising new approach to CG in machine learning, demonstrating potential in empirical tests. However, there are areas where improvements are needed, particularly in terms of clarity and detail in the methodology.

**Strengths And Weaknesses:**

Pros:
1. The paper addresses a significant and relatively unexplored aspect of machine learning: CG, highlighting its practical implications.

2. The introduction of CFA is innovative and shows promise, as evidenced by empirical evaluations.

3. The creation of CG-Bench marks a valuable contribution, providing a resourceful benchmark for future research in this area.

Cons:
1. While the theory presented in the paper aims to learn features with robustness and interpretability, there are doubts about the generalizability of the results. Figure 5, although impressive, may represent a selectively favorable scenario.

2. The methods to maintain orthogonality between W_1 and W_2 during training, and the integration of constraints in Equation (2) into the overall optimization objective, are not clearly explained. More detail on the optimization process and the impact of the orthogonal constraint on loss optimization would be beneficial. Visualizations of loss curves with orthogonal contrast ablations would enhance clarity.

3. There are typographical errors in Table 1, such as "943.07" or "54.33.2".

4. The overall improvements presented in the paper, while positive, are not markedly significant.

---

> ### Author Response · Authors · 2024-03-28
> **Author Response to Reviewer kZdn**
>
> ### **Weakness 1 (Quantitative Evaluation beyond the Visualization of Fig. 5)**
>
> We are glad that you find our Fig. 5 impressive, and we understand your concern that it may represent a selectively favorable scenario. To address your concern, we conducted a quantitative evaluation of the feature orthogonality for the pre-trained model and our CFA-finetuned model on the DomainNet dataset, using the same setup as in Figure 5. Specifically, for each domain-class combination, we computed the empirical mean of data points in the feature space as the centroid of this domain-class combination. Then, we fit two linear classifiers, $W_1$ for domain labels and $W_2$ for class labels, to all the centroids (essentially, we optimized with Eq. 1 as the objective without the orthogonality constraint for classifiers). Finally, we computed $|W_1^T W_2|_F^2$ as a feature orthogonality metric to assess how orthogonal the learned features are (the lower, the better). The feature orthogonality metrics for the pretrained and SFT-finetuned CLIP models are presented below, and it is obvious that CFA enjoys much stronger orthogonality than the pretrained model.
>
> | Model | Feature Orthogonality |
> |--------|-------------------|
> | Pretrain    | 8.23    |
> |     CFA  | **2.84**|
>
>
> ### **Weakness 2 (Impact of the Orthogonality Constraint on Loss Optimization)**
>
> Equation 2 represents a hard orthogonality constraint on the two classifiers, $W_1$ and $W_2$. In practice, as detailed in Sec. 3.2, we use a soft version of the constraint by introducing a regularization term $|W_1^T W_2|_F^2$ to the training objective. This is a common implementation for orthogonality constraints and is also adopted in other works, such as [1]. To illustrate how this orthogonality regularization impacts the optimization process, we conducted an ablation study on Stage-1 of our CFA on the DomainNet dataset. Specifically, we optimized the objective in Equation 1 with an orthogonality regularization for 20 epochs on the DomainNet dataset using different orthogonal regularization coefficients. We visualized the results in Fig. 7 in the Appendix of the updated manuscript, which demonstrates that the orthogonality regularization is effective for obtaining orthogonal heads $W_1$ and $W_2$ to be used in Stage-2.
>
> ### **Weakness-3 (Typo)**
>
> We addressed the typo in Table 1 and updated the manuscript accordingly. We are grateful for your careful review.
>
>
>
> ### **Weakness 4 (Performance Gain)**
>
> We appreciate your concern regarding the empirical improvement of our algorithm and understand that you believe the performance gain is not substantial enough. In the submission, we included a discussion on this matter in Appendix C.2, where we explain the reasons behind the seemingly modest performance improvement. For instance, we elaborated on the issues of mislabeling and ambiguous labels that we encountered in the datasets used in our experiments. These factors can potentially limit the observable performance gain. We highly encourage you to review Appendix C.2, as it provides valuable insights into the challenges we faced and how they may have impacted the results. We hope that this additional information will help address some of your concerns and provide a clearer understanding of our algorithm's performance in the context of these datasets.
>
> ## References
> [1] Zhang et al. AdaLoRA: Adaptive Budget Allocation for Parameter-Efficient Fine-Tuning. ICLR 2024

---

### Review · Reviewer_L4Jk · 2024-03-13

**Summary Of Contributions:**

This paper proposes a method to achieve compositional generalization across domains and item classes via a two stage learning process.
First, orthogonal domain and classes discriminators are learned given a fixed set of features and then in the second stage, a new set of features are learned given fixed class and domain discriminators.
The paper shows some improvement of using this method over reasonable baselines.

**Audience:**

Yes

**Claims And Evidence:**

Yes

**Requested Changes:**

1. The clip-CFA row in Table 1 has some incorrectly formatted #s in the first 2 columns
2. Since the paper is currently under the 12 page limit, the Ablations could have been included in the main body of the paper instead of the appendix.
3. **[Requested Experiment]** It would be good to have an ablation with a single stage version of the whole fine-tuning process to confirm that the 2 stage process is indeed needed though it is more complex and has extra overhead. Specifically,  having a single stage of training where the body parameters and heads are all jointly trained ---> with and without the orthogonality constraints on the heads.

**Strengths And Weaknesses:**

## Strength
1. Paper is well written and motivations are clearly stated. Good problem setup with helpful visualizations that clearly convey the message of the paper
2. Method is relatively simple but effective as results from Table 1 demonstrate
3. Experiments using Clip to generate domain labels show compelling performance and an ability to leverage the approach even in settings without domain labels.


## Weaknesses
1. The benchmark is a bit overstated as a contribution since 2/3 of the sub-tasks are basically wholesale lifting of existing datasets with no real modification.
2. The theoretical guarantees given are with respect to training data but say nothing about  whether the aligned featurizer also exhibits the compositional property for data not in the training set.

## Questions :
1. How do you decide what numbers to bold in Table 1 ?  On FMoW, the performance of Reweight-E with clip (41.8) is better than your method (41.6) but your method's result is instead bolded.
2. Any idea hypotheses about why WISE tends to hurt performance on the FMoW dataset ?

---

> ### Author Response · Authors · 2024-03-28
> **Author Response to Reviewer L4Jk**
>
> ### **Weakness 2 (Generalization to Unseen Domain-Class Combinations)**
> Our Theorem 1 is indeed only about training data. For domain-class combinations that do not appear in the training data, our current theoretical framework cannot guarantee that their features will conform to the compositional structure. The reason for that is that our framework adopts the Unconstrained Feature Model (UFM) from the neural collapse literature, which essentially assumes the featurizer is sufficiently powerful and easily optimizable; however, this assumption implicitly allows the featurizer to brute-forcely memorize all training data (i.e., overfitting) without any generalization to unseen data (e.g., the featurizer can learn simplex structure for features as dictated by neural collapse [1], even for randomly labeled data). Hence, if we want to guarantee generalization to unseen domain-class combinations, we have to remove this assumption and consider limited-capacity featurizer classes. For instance, if we consider that the ground-truth data are generated with a linear feature mapping from a latent compositional feature structure (similar to the causal model considered in [2][3][4]), then we can assume the featurizer to be a linear model and leverage technical skills from work such as [2][3][4] to prove that our model can learn this compositional feature structure up to an invertible linear mapping, which guarantees generalization to unseen domain-class combinations. However, for general non-linear models, it would be much more difficult to prove the generalization (e.g., [2] proves an impossibility theorem for IRM's learning of invariant features with non-linear models), as larger-capacity models can more easily overfit to the training data. In summary, if we want to prove generalization, we need stricter assumptions on the data generative process and also restrict the study to weaker model classes (e.g., linear models). This is totally achievable, but it is not the focus of this work, as we want our theoretical framework to be compatible with modern powerful model classes such as ConvNets and Transformers.
>
> ### **Question 2 (Why WiSE hurts performance sometimes)**
>
> WiSE can generally improve the OOD performance of CLIP and other vision models. However, the optimal WiSE coefficient is not known in advance. In our work, we use a WiSE interpolation coefficient of 0.5 because it is unreasonable to assume we can use the OOD accuracy to find the optimal coefficient (as the unseen domain-class combinations are not available to us in practice). As shown in previous works such as [5] (e.g., Fig 2 of [5]), for CLIP on the FMoW dataset, a coefficient of 0.5 may not give better OOD performance, as the optimal coefficient is around 0.8. This is also the reason for the performance drop in our case for WiSE with a coefficient of 0.5.
>
> ### **Requested Experiment**
>
> | Method | $\lambda_{ortho}$ | ID Acc | OOD Acc |
> |--------|-------------------|--------|---------|
> | 1-stage     | 0                 | 81.8   | 7.0     |
> |     1-stage   | 0.1               | 81.8   | 7.1     |
> |      1-stage  | 1                 | 81.8   | 7.1     |
> | **2-stage** | -                 | 81.6   | 7.3     |
>
>
>
> We conducted an ablation study as requested. Specifically, we jointly finetuned the pretrained encoder and two heads with different orthogonal regularization coefficients on the DomainNet dataset, and we denoted this as "1-stage" in the following table, in contrast to our original 2-stage training method. From the results, we can see that our 2-stage pipeline results in better OOD performance. This result is also aligned with findings in [6], which observe that 2-stage training (linear probing followed by full finetuning) can yield better OOD performance than the vanilla 1-stage full fine-tuning.
>
>
> ### **Manuscript Revision**
>
> We are grateful for your careful review of the paper, and we have revised our manuscript to address the bolding and formatting issues of Table 1 that you pointed out. Note: now we use bold numbers for the highest accuracy and those within a range of 0.2.
>
> ## References
>
> [1] Kothapalli. Neural Collapse: A Review on Modelling Principles and Generalization. TMLR 2023
>
> [2] Rosenfeld et al. The Risks of Invariant Risk Minimization. ICLR 2021.
>
> [3] Wang et al. Provable Domain Generalization via Invariant-Feature Subspace Recovery. ICML 2022
>
> [4] Wang et al. Instance-Dependent Partial Label Learning with Identifiable Causal Representations. NeurIPS workshop, 2023
>
> [5] Goyal et al. Finetune like you pretrain: Improved finetuning of zero-shot vision models. CVPR 2023
>
> [6] Kumar et al. Fine-Tuning can Distort Pretrained Features and Underperform Out-of-Distribution. ICLR 2022

---

> > ### Comment · Reviewer_L4Jk · 2024-04-15
> > **Response to rebuttal**
> >
> > Thank you for your response to my questions.
> > I am satisfied with the responses.
> > Please update the manuscript to also contain a discussion limitation of the theory with respect to generalization error.

---

### Decision · Action_Editor_iXPp · 2024-04-24

**Recommendation:** Accept as is

**Comment:**

The reviewers found the paper well-written with motivations clearly stated (Reviewer L4Jk), the proposed method is simple (Reviewer L4Jk, Reviewer kFXZ) and effective (Reviewer L4Jk) and “shows promise, as evidenced by empirical evaluations” (Reviewer kZdn). The paper addresses a “significant and relatively unexplored aspect of machine learning” (Reviewer kZdn). While Reviewer kFXZ is concerned about limiting assumptions made that aren't met in (all) real-world applications, the authors have made claims within the scope of that setting and have clearly outlined the assumptions made -- e.g. "Empirically, we find that if the learned features (i.e., the outputs of the last hidden layer) conform to a compositional structure where the subspace of domain related features is orthogonal to that of class-related features, the corresponding model can generalize across unknown domain-class pairs."


During the rebuttal, the authors have made clarifications, expanded on the discussion of limitations and conducted an additional ablation experiment to verify the necessity of the two-stage design of their method.


I encourage the authors to add the additional discussion of limitation to the revised manuscript as well.

**Audience:**

Yes, this paper is on the topic of compositional generalization which is of interest to the TMLR community. As the authors point out, some problems that are often posed as Domain Generalization problems can actually also be seen as CG. Also, domain generalization may make a different set of unrealistic assumptions, e.g. that data from a set of diverse domains is available at training time. It seems that CG and domain generalization may each be most applicable in different applications and both are relevant and scientifically interesting.

**Claims And Evidence:**

This paper studies Compositional Generalization (CG) where a model is tasked to generalize to a previously unseen class-domain combinations, albeit having seen different combinations of the same classes / domains before. The authors make a simplifying assumption that these two axes (class labels and domains) are causally independent from each other. They propose a new method, inspired by previous work, for tackling CG. Their method has two stages: the first learns two linear heads on a pretrained encoder (to predict class and domain labels) while keeping the encoder frozen, wile the second freezes those newly learned heads and finetunes the encoder to align with them. They also put together a suite of CG tasks (by heavily reuisng existing datasets, in some cases unchanged) and conduct an empirical investigation of their method and relevant baselines, including ablation studies and extensions to the case where domain labels aren't available. They claim that their method outperfoms previous works on their benchmark, which they do show empirically though, as the reviewers pointed out, the proposed method is only marginally better. They also theoretically substantiate the claim that their method encourages ‘compositional feature learning’. Overall, the reviewers did not point out issues of unsubstantiated claims in the paper. While Reviewer kFXZ is concerned about limiting assumptions made that aren't met in (all) real-world applications, the authors have made claims within the scope of that setting and have clearly outlined the assumptions made -- e.g. "Empirically, we find that if the learned features (i.e., the outputs of the last hidden layer) conform to a compositional structure where the subspace of domain related features is orthogonal to that of class-related features, the corresponding model can generalize across unknown domain-class pairs."